# Peer review of "The Effect of Hydrogen Gas and Water Vapor in Catalytic Chemical Vapor Deposition on the Structure of Vertically Aligned Carbon Nanotubes"

_materials, 2025, doi:10.3390/ma18235309_

Round 1
Reviewer 1 Report
Comments and Suggestions for Authors
This manuscript shows VA-CNT synthesis on Si substrates, investing the effect of water and H2 supply during CNT growth for FeCo catalysts. There have been a large amount of papers in this field, but this manuscript does not give any significant information on VA-CNT synthesis. Also, important information are lack, such as the structure CNT, the size of catalyst particles and composition of catalysts . So, this manuscript is not suitable for publication.
Author Response
See attached Word file.

Reviewer 2 Report
Comments and Suggestions for Authors
Review “Effect of synthesis parameters in catalytic chemical vapor deposition on the structure of vertically aligned carbon nanotubes”
The process of formation of CNT on the substrates described in lines 150-157 is taken directly from the literature findings. The authors presented no evidence that exactly the same processes have taken place under their reaction conditions.
The process of CNT formation has been studied in detail for many years, thus it is not clear what the novelty of this work is. Other similar works should be added in the introduction to better explain the novelty of this study.
Figure 3 needs to be better formatted.
How do the authors explain the discrepancy between their results and those reported in the literature (line 177?)
The SEM images should be presented side by side. It is not necessary to present the results for all ratios of catalysts for each of the parameters studied. The manuscript should be revised so that paragraph 3.4 becomes paragraph 3.2. In paragraph 3.1, the authors should discuss the influence of the catalyst ratio on the properties of CNT and conclude that the same trends were observed regardless of the catalyst ratio used in studies of other parameters. The remaining SEM images can be included in the supplementary material.
Do the authors have any evidence that the water vapour “oxidized” CNTs? It should be added to the manuscript.
Are the results obtained here without hydrogen different from the literature reports?
There is not enough data in this manuscript. The authors should discuss the yield of carbon nanotubes obtained with different ratios of catalysts. Additionally, the results should be compared to those reported in the literature. Some characteristics of the surface chemistry should also be added since the authors claim that the nanotubes were oxidized, but no evidence of this was presented. The thermal stability of these nanotubes would also be an important characteristic to study.
Author Response
See attached Word file.

Reviewer 3 Report
Comments and Suggestions for Authors
Since their discovery, substantial research advancements have been made in the field of carbon nanotubes (CNTs). The high interest to this material is due to CNTs exceptional properties—including outstanding tensile strength and superior electrical conductivity. The excellent physical and chemical characteristics of CNTs enable their use across a wide range of fields. Applications include composite material development in industries like construction, energy, and automotive, as well as environmental technologies such as gas adsorption, sensing, and fuel cells. Additionally, CNTs serve as critical elements in supercapacitors and electrodes. Supercapacitors are particularly promising as they bridge the gap between electrolytic capacitors and rechargeable batteries. A distinct category of carbon nanotubes, known in the literature as vertically aligned carbon nanotubes (VACNTs) or carbon nanotube forests, is characterized by their well-oriented structure. While they share the fundamental properties of conventional CNTs, their ordered arrangement confers superior electrical and thermal conductivity. Consequently, their most promising applications are anticipated in electronic and microelectronic devices—such as high-capacity capacitors, solar cells, photodiodes, electrodes, and gas sensors—as well as in high-performance composites. A key differentiator from conventional CNTs is their production method: VACNTs can only be synthesized through catalytic chemical vapor deposition (CCVD), which involves the thermal decomposition of hydrocarbons or other carbon-containing precursors. In the present work, VACNTs were synthesized on silicon substrates to examine the effects of water vapor and hydrogen gas on their growth dynamics and structural properties. Additionally, the role of catalyst composition was assessed by employing a bimetallic iron-cobalt (Fe-Co) system.
The authors conducted a systematic study, varying three key parameters in three different combinations (Hâ‚‚+Hâ‚‚O, Hâ‚‚ only, Hâ‚‚O only) for five catalyst compositions. This approach allows for a multifaceted comparative analysis.
The SEM images clearly demonstrate the differences in morphology and alignment of the tubes under different synthesis conditions, fully confirming the conclusions made in the text.
A significant practical implication of this study is the conclusion that while hydrogen is indispensable for VACNT growth, water vapor is not required to achieve high-quality nanotube arrays. This finding streamlines the synthesis protocol, offering a path toward reduced operational complexity and lower production costs. The authors not only report their findings but also interpret them through the lens of catalytic mechanisms, specifically citing hydrogen's role in catalyst regeneration and water's efficacy in amorphous carbon removing. This analytical approach provides a deeper understanding of the underlying processes.
The work is not without its flaws. Some notes to the work can be made.
- The paper presents the average values of the heights of the structures obtained, but does not indicate the standard deviations or the number of measurements taken for each point. Statistics SHOULD be added. This would increase the reliability of the results.
- A more detailed discussion IS NEEDED to address the contradiction the authors note between their data and prior literature. It is possible that this discrepancy arises from the unique catalyst morphology and distribution resulting from the PLD application method, as opposed to more conventional techniques.
- Although SEM and Raman spectroscopy are essential methods, the addition of transmission electron microscopy (TEM) data would allow for accurate assessment of tube diameter, number of walls, and the state of catalytic particles at the tips. Specific surface area analysis (BET) could quantitatively characterize surface development.
- To fully demonstrate the potential of the materials obtained, it would be extremely useful to conduct simple functional tests, such as measuring the electrical conductivity of the VACNT arrays obtained, which is directly related to their quality and orientation.
This manuscript is in principal suitable for publication in its current form. The authors are nevertheless strongly encouraged to address the aforementioned points regarding statistical height data and a more detailed discussion of conflicting literature. As well as more experimental efforts are recommended in their future work to further solidify the impact of their findings.
Author Response
See attached Word file.

Reviewer 4 Report
Comments and Suggestions for Authors
This study systematically investigates the influence of water vapor and hydrogen gas on the structure and properties of vertically aligned carbon nanotubes (VACNTs) grown via catalytic chemical vapor deposition (CCVD). The experimental design is rational, the methodology is well-documented, and the findings hold scientific value and practical relevance. The manuscript is clearly structured, and the conclusions are supported by the data. The key conclusion that hydrogen is essential for VACNT synthesis, while water vapor improves quality but is not strictly necessary, provides valuable guidance for optimizing synthesis protocols. The experimental design is reasonable and the data analysis is detailed, which has certain academic value.
But there are some questions, that must be solved before it is considered for publication.
- The manuscript refers to references [34,35] for details on the PLD catalyst deposition. However, key parameters (e.g., laser energy, number of pulses, deposition atmosphere) should be briefly summarized in the main text to enhance reproducibility.
- The SEM images in Figures 2, 3, and 4 are of low resolution, and scale bars are missing or unclear. Higher-resolution images with consistently placed and clearly labeled scale bars are recommended.
- The anomalous result of the shortest VACNTs being formed with the 1:1 Fe:Co catalyst ratio warrants a more in-depth discussion of potential reasons (e.g., catalyst particle size, oxidation state).
- The observed trend in Raman I_D/I_G ratios seems inconsistent with the SEM-based conclusion that CNT quality is better at the top. This discrepancy should be discussed (considering factors like laser penetration depth or sample inhomogeneity).
- Some sentences are overly long or grammatically awkward (e.g., "the gentle oxidative environment that would have reduced the height..."). Minor language polishing for clarity and conciseness is advised.
- The reference formatting is inconsistent; some entries lack journal names, volume/issue numbers, or complete details. Please ensure uniformity according to the journal's style guide.
Comments on the Quality of English Language
But there are some questions, that must be solved before it is considered for publication.
- The manuscript refers to references [34,35] for details on the PLD catalyst deposition. However, key parameters (e.g., laser energy, number of pulses, deposition atmosphere) should be briefly summarized in the main text to enhance reproducibility.
- The SEM images in Figures 2, 3, and 4 are of low resolution, and scale bars are missing or unclear. Higher-resolution images with consistently placed and clearly labeled scale bars are recommended.
- The anomalous result of the shortest VACNTs being formed with the 1:1 Fe:Co catalyst ratio warrants a more in-depth discussion of potential reasons (e.g., catalyst particle size, oxidation state).
- The observed trend in Raman I_D/I_G ratios seems inconsistent with the SEM-based conclusion that CNT quality is better at the top. This discrepancy should be discussed (considering factors like laser penetration depth or sample inhomogeneity).
- Some sentences are overly long or grammatically awkward (e.g., "the gentle oxidative environment that would have reduced the height..."). Minor language polishing for clarity and conciseness is advised.
- The reference formatting is inconsistent; some entries lack journal names, volume/issue numbers, or complete details. Please ensure uniformity according to the journal's style guide.
Author Response
See attached Word file.

Round 2
Reviewer 2 Report
Comments and Suggestions for Authors
The manuscript has not been sufficiently improved, and it still needs further revision. The Reviewer's doubts were still not sufficiently addressed.
The Reviewer asked about the novelty of the work in the first revision. In their introduction, the authors claim that there are literature reports about the influence of various parameters, such as catalyst and support layer quality, layer formation conditions, synthesis time, gas flow rates, and water vapor quantity (introduction, lines 59-70). It is still not clear what the novelty of this work is if the synthesis of VACNT on a silicon substrate has been studied, and the influence of water and gas has been studied. In their response after the first revision,, the authors claimed that the use of the PLD method is not so widespread, but a simple Google search returned some results, see for example:
https://www.sciencedirect.com/science/article/pii/S0040609020304041
The materials list (paragraph 2.1) should be put in a Table or in a formatted way.
Figure 1 has a very poor quality.
In response to the reviewer’s question about the discrepancy between the results obtained by the authors and the published data, the authors explained that the quality of the product depends on the location of the laboratory. In this sense, is it even worth studying the effect of different parameters on the process that can not be repeated in another laboratory?
The manuscript continues to have very little data; the authors present SEM images of the materials prepared with different ratios of catalyst to show the influence of the studied parameters, but the conclusions drawn are the same for all of the catalyst compositions used. Therefore, it is not necessary to show all the samples for all of the parameters studied. This comment was already made in my previous revision.
The authors do not have any characterisations of these materials apart from SEM and Raman, but they discuss the oxidation of nanotubes without presenting any results to prove their conclusions. Citing their previous works to support their conclusions may give the impression that this work repeats previous findings if all the characteristics are the same as previously reported. And how are the characteristics the same if the process is so sensitive? More tests were recommended, but the authors replied that they did not have sufficient time to do them. I am sure extra time can be requested from the editor to run some more tests.
Reviewer 4 Report
Comments and Suggestions for Authors
According to the suggestion before, the authors have revised this article including the language such as grammar and format, interpunction, the authors also gave the reasonable expression about the questions.
Comments on the Quality of English LanguageAccording to the suggestion before, the authors have revised this article including the language such as grammar and format, interpunction, the authors also gave the reasonable expression about the questions.
Round 3
Reviewer 2 Report
Comments and Suggestions for Authors
The manuscript has been significantly improved and is ready for publication.
Author Response
Thank you for your positive feedback!